# The Lyon Consensus Criteria for GERD Diagnosis in a Greek Population: The Clinical Impact and Changes in GERD Diagnosis in a Real-World, Retrospective Study

**DOI:** 10.3390/jcm11185383

**Published:** 2022-09-14

**Authors:** Theodoros Voulgaris, Vasileios Lekakis, Afroditi Orfanidou, Jiannis Vlachogiannakos, Dimitrios Kamberoglou, George Papatheodoridis, George Karamanolis

**Affiliations:** Department of Gastroenterology, School of Medicine, National and Kapodistrian University of Athens, Agiou Thoma 17, Goudi, 11527 Athens, Greece

**Keywords:** pHmetry impedance, gastroesophageal reflux disease, acid exposure time, mean nocturnal basal impedance, proton pump inhibitors

## Abstract

(1) Introduction/aim: Gastroesophageal reflux disease (GERD) affects 8–33% globally. The gold standard examination technique in diagnosing GERD is 24 h pHmetry ± impedance. Recently, new diagnostic criteria were introduced by the Lyon Consensus for GERD diagnosis. Our aim was to investigate the diagnostic yield of pHmetry + impedance using the Lyon Consensus criteria in a real-world study. (2) Patients and methods: Our study included 249 consecutive patients (M/F: 120/129, mean age 50 ± 15 years) who underwent 24 h pH+ impedance monitoring in our department, during a 5-year period. Epidemiological, endoscopic, clinical, and 24 h pH+ impedance data were retrospectively collected. (3) Results: Typical GERD symptoms were reported by 140/249 (56.2%) patients, whereas 99/249 (39.6%) patients reported various extraesophageal symptoms. Endoscopic findings supportive of GERD based on the Lyon Consensus were present in 42/185 (22.7%). An AET value of >6% was observed in 60/249 (24.1%). GERD diagnosis according to the Lyon Consensus criteria was set in 63/249 (25.3%) patients; a rate significantly lower than that observed by implementing the older criteria (32.1%), *p* < 0.001. In the multivariate analysis, the existence of endoscopic findings supportive of GERD diagnosis as defined by the Lyon Consensus (*p* = 0.036), a De Meester score of over 14.7, and the presence of typical GERD symptoms were correlated to GERD diagnosis (*p* < 0.001, respectively) using the criteria defined for pH–impedance monitoring. (4) Conclusions: Changes in the diagnostic criteria concerning the 24 h pH–impedance monitoring of GERD based on the Lyon Consensus led to a conclusive GERD diagnosis in approximately 25% of the patients. This rate of GERD diagnosis is reduced in comparison to the one confirmed with the use of previously established criteria.

## 1. Introduction

Gastroesophageal reflux disease (GERD) is one of the commonest gastroenterological diseases. Globally, it affects 8–33% of the population without differences among sex, and it can be observed among all age groups [1]. The clinical manifestations of GERD are variable, including not only typical symptoms such as retrosternal burning chest pain and regurgitations but also multiple non-typical and extraesophageal symptoms. Adding the absence of simple laboratory and radiological studies, confirmation of GERD diagnosis is still a challenging project [2].

Due to the high frequency of GERD in the general population, most of the clinical guidelines accept the simple usage of medical history for diagnosis [3]. Unfortunately, the sensitivity and specificity of patients’ medical history in diagnosing GERD, even if it comes from an expert in the field, do not exceed 70%, respectively [4]. Multiple questionnaires have tried to overcome this diagnostic barrier, but none has consistently led to acceptable diagnostic accuracy [5]. Aside from patients’ medical history and symptoms, another well-recognized method in diagnosing GERD is patients’ response to a short trial of proton pump inhibitors (PPIs). A clinical response to a 15-day trial with once-daily PPI treatment can lead to the diagnosis of GERD, though still with a sensitivity of 71% and a specificity of 44%, in comparison to the combination of upper GI endoscopy and pHmetry [6]. It should be underlined that in up to one-third of those patients without GERD evidence, a short PPI trial may ameliorate their symptoms [7]. As mentioned above, the diagnostic accuracy of patients’ medical history and response to a PPI trial is suboptimal, but their usage has still been accepted due to cost-effectiveness [8,9,10].

In those patients who do not respond to treatment, in patients with diagnostic uncertainty, or in whom an invasive treatment has been proposed, the diagnosis of gastroesophageal reflux should be documented using the gold standard examination technique in diagnosing GERD: 24 h pHmetry ± impedance. The main metric of pHmetry used for the diagnosis of gastroesophageal reflux is the percentage of patients’ esophageal acid exposure time (AET) [11], even though a specific AET cut-off has not been defined. Most published studies used a cut-off of 4.2% in order to diagnose GERD in those patients undertaking the examination without being treated with PPIs [12,13]. The recent Lyon Consensus has defined specific cut-offs for all the metrics in pHmetry ± impedance, which conclusively establish the presence of GERD and also define the characteristics that rule out GERD. According to the Lyon Consensus, conclusive evidence for GERD is provided by either endoscopy or pHmetry ± impedance [14]. Thus, patients with grade C or D esophagitis, Barrett’s esophagus, and benign esophageal stenosis can be diagnosed with GERD based solely on endoscopic findings, whereas esophagitis grade A or B at endoscopy are considered borderline evidence. Using pHmetry ± impedance, the AET cut-off for GERD diagnosis has been defined as 6%, whereas AET < 4% is considered normal. An AET value between 4% and 6% is considered a grey zone, and therefore, supportive findings could be added in order to confirm or refute the diagnosis of GERD. Due to the recent publication of the Lyon criteria, the consequences in the rates of GERD diagnosis have not yet been evaluated in large-scale studies. Subsequently, their correlation to the current methods of GERD diagnosis, used commonly by clinicians and adopted from national societies such as PPI responsiveness and patients’ medical history is also not yet validated.

The aim of our study was to investigate the diagnostic yield of pHmetry + impedance using the Lyon Consensus criteria in a real-world study and to evaluate possible changes in the rate of GERD diagnosis with the adaptation of those new diagnostic criteria for patients partially or not at all responsive to PPIs without conclusive endoscopic evidence of GERD and patients with a previous GERD diagnosis and a complete response to PPIs who were candidates for surgical treatments.

## 2. Materials and Methods

### 2.1. Patients

Data from consecutive patients who underwent 24 h pH+ impedance monitoring in the laboratory for esophageal disorders of the Gastroenterology Academic Department of National and Kapodistrian University of Athens during a five-year period (6/2016–6/2021) were retrospectively collected. We included patients with long-lasting (symptoms not less than 6 months), typical (caustic chest pain, regurgitation), or atypical esophageal symptoms (non-caustic retrosternal pain, dysphagia) that were unresponsive or partially responsive to PPIs, as well as those responsive to PPIs but requiring 24 h pH–impedance monitoring in order to confirm GERD diagnosis in the setting of pre-surgical control for anti-reflux surgery. The following extraesophageal manifestations of GERD were considered: chronic cough, laryngitis, and asthma. Patients with recent (previous 3 months) use of any medications that could lead to the development of symptoms, as well as patients with a history of surgery that could result in abnormal gastrointestinal motility or reflux itself, were excluded. All patients previously received PPI therapy, with either single or double doses, for at least 8 weeks. Complete PPI response was defined as the disappearance of symptoms after the institution of PPI treatment. Partial response was defined as an amelioration of patient symptoms but without the complete disappearance of them, and no response was defined as no change to patient symptoms. The examination was undertaken off PPI in the total cohort. In total, 44 tracings of 24 h pH–impedance monitoring were prospectively analyzed twice separately. Only pH–impedance studies in which the probe remained in place for at least 16 h were considered valid for analysis. The artifacts recognized by the operator were manually excluded from the analysis. The data were analyzed once by using the older diagnostic criteria (AET > 4.2%) and once more by implementing the new proposed by the Lyon Consensus diagnostic criteria. For each patient, the epidemiological, endoscopic, and clinical data were analyzed.

The study was conducted in compliance with the ethics principles for medical research involving human subjects as stated in the Declaration of Helsinki of the World Medical Association.

### 2.2. 24 h pH + Impedance Monitoring

The patients were required to fast for at least 8 h before the pH ± impedance monitoring. The pH + impedance probe consisted of a polyurethane catheter (VERSAFLEX Z, GIVEN IMAGING) that included six impedance segments (each segment was 2 cm long) and one pH-measuring electrode. The configuration of this catheter enabled the recording of changes in the intraluminal impedance at 3, 5, 7, 9, 15, and 17 cm above the LES. Additionally, the pH was monitored at 5 cm above the LES. The pH ± impedance probe was inserted trans-nasally, and the distal pH probe was positioned 5 cm above the LES as identified by using high-resolution esophageal manometry. The data from the impedance channels and the pH electrodes were transmitted at a frequency of 50 Hz and stored on a portable data recorder (Ohmega Portable 24 h Impedance and PH system, MMS, Enschede, The Netherlands). Data recording was concluded after 24 h when the patients returned to the esophageal laboratory for probe removal. The patients were instructed to complete a diary that included indications of the beginning and ending times of meals and changes in body position and were asked to report in the same diary the exact time whenever they experienced reflux symptoms as well as the type of symptom. Acid exposure time (AET) was calculated as the percentage of time during which the pH was below 4 according to the esophageal pH sensor, and AETs of 4.2% or 6% and greater were designated as abnormal thresholds.

The number of reflux episodes (NREs), reflux–symptom association (symptom index (SI) and symptom association probability (SAP)) as well as the mean nocturnal baseline impedance (MNBI) was also calculated. Post-reflux swallow-induced peristaltic wave (PSPW), a novel PHmetry index that may in the future add to GERD diagnosis, was not measured in our study due to a lack of resources. (14) A reflux episode was identified by a 50% decrease in impedance lasting for at least 4 sec each in distal 2 impedance channels with retrograde propagation [15]. NREs were manually reviewed. The Lyon Consensus suggests that NREs > 80/24 h are definitively abnormal, whereas NREs < 40/24 h are normal, and intermediate values are inconclusive. The symptoms were considered related to reflux events if they occurred within 2 min after the reflux events. The symptom index (SI) and symptom association probability (SAP) were calculated and designated as positive when SI > 50% or SAP > 95%. The assessed symptoms were heartburn, regurgitation, chest pain, cough, or belching [14]. MNBI was calculated as a mean of 3 different nocturnal periods’ baseline impedance values at 3 cm above LES. Values < 2292 Ohms (Ω) were considered abnormal [16].

### 2.3. GERD Definition

A conclusive GERD diagnosis was made when an abnormal AET value (either >4.2% or >6%) was calculated. Based on the Lyon Consensus, additional pH–impedance metrics suggestive of GERD diagnosis were proposed: When AET was considered inconclusive (between 4–6%), adjunctive metrics such as abnormal NRE, positive SI or SAP, and low MNBI were used in order to establish GERD diagnosis.

### 2.4. Statistical Analysis

Statistical analysis was performed using SPSS V23 (SPSS software; SPSS Inc, Chicago, IL, USA). Data are expressed as frequencies, mean ± SD, or median (interquartile range, IQR), as appropriate. Quantitative variables were compared between groups with Student’s *t*-test or Mann–Whitney test for normally distributed and non-normally distributed variables, respectively. Qualitative variables were compared with the chi-squared test or Fisher’s exact test, as appropriate. Multivariate logistic regression analysis models were used to identify the independent, significant, and predictive factors of a poor dichotomous outcome. Only the parameters with a significant or a trend for significant association (*p* < 0.10) with the dependent variable in the univariate analysis were included in the multivariate analysis models. All the tests were two-sided, and *p* values < 0.05 were considered to be significant.

## 3. Results

### 3.1. Study Population

A total of 249 patients (129 female; mean age 50 ± 15 years, range 18–86) fulfilled the inclusion criteria were evaluated. The patients’ mean BMI was 24.8 ± 3.0 kg/cm^2^. The patients’ symptoms are summarized in Table 1; in total, 140/249 (56.2%) reported typical GERD symptoms (regurgitations, retrosternal caustic pain, and retrosternal pain), whereas 99/249 (39.6%) reported various extraesophageal symptoms (asthma, laryngitis, and chronic cough). The majority (137/249, 55%) of the patients reported no amelioration of their symptoms when previously treated with PPIs, while 79/249 (31.7%) of the patients reported partial response. Forty-three patients (17.3%) underwent pHmetry + impedance in order to confirm pathological reflux prior to anti-reflux surgery.

The endoscopic data were available in 172/249 patients. Esophagitis was present in 25/172 (14.5%) patients (grade A: 13/172, B: 12/172), whereas hiatal hernia was endoscopically found in 24/172 (13.8%) patients. In total, endoscopic findings supportive of GERD based on the Lyon Consensus were present in 42/172 (24.4%).

### 3.2. pH–Impedance Monitoring Results

Among the total study cohort, 60/249 (24.1%) patients had an AET value of >6%, 22/249 (8.9%) between 4% and 6%, and 147/249 (59%) had <4%. The mean number of reflux episodes was 32 ± 27 (min 0–max 198), while only 12/249 (4.8%) patients had over >80 reflux episodes. The mean De Meester score was 16.3 ± 25 (min 0.2–max 268.5), and 74/249 (29.7%) patients had a De Meester score > 14.7. SI and SAP were calculated in 172/249 patients, because 77 patients did not report any symptoms during the study. Abnormal MNBI was observed in 38/249 (15.3%) patients (Table 2).

### 3.3. GERD Diagnosis Based on AEt Alone

Figure 1 shows the rate of GERD diagnosis using different pathological AET cut-offs. When adopting the older AET cut-off (4.2%) for diagnosing GERD in the total cohort, GERD diagnosis was made in 80/249 (32.1%) patients. Using the AET cut-off proposed by the Lyon Consensus, a definite GERD diagnosis was made in 60/249 (24.1%). Thus, the rate of GERD diagnosis based only on pathological AET was significantly lower contrary to when using the older AET criteria (24.1% vs. 32.1%, *p* < 0.001) (Figure 1).

### 3.4. GERD Diagnosis Based on Supportive PH + Impedance Metrics

An inconclusive GERD diagnosis with an AET value between 4% and 6% was observed in 22/249 (8.8%) patients. Among those, supportive pH + impedance variables adding confidence to the presence of GERD were found in 3/22 (13.7%) patients: One patient had abnormal NRE, and two patients demonstrated abnormal MNBI. None of the patients in the inconclusive GERD group showed positive SI and/or SAP.

In total, when implementing the pH + impedance monitoring criteria of the Lyon Consensus, GERD diagnosis was set in 63/249 (25.3%) patients. Even with the addition of supportive abnormal pH–impedance metrics, the rate of GERD diagnosis was significantly lower than the rate before the introduction of the new Lyon Consensus criteria (25.3% vs. 32.1%, *p* < 0.001).

### 3.5. Prognostic Factors of GERD Diagnosis When Using the Lyon Consensus Criteria for pH + Impedance Monitoring

Patients’ BMI was correlated to GERD diagnosis. Those patients diagnosed with GERD by implementing the Lyon Consensus criteria had a statistically significant greater BMI (25.97 kg/cm^2^ vs. 24.55 kg/cm^2^, *p* = 0.015). Moreover, male patients showed a greater rate of being diagnosed with GERD (37/120, 30.8% vs. 24/129, 18.6%, *p* = 0.028), while patients with GERD diagnosis tended to be younger (51 vs. 47 years old, *p* = 0.097). The presence of typical GERD symptoms was also significantly correlated to GERD diagnosis in comparison to atypical symptoms (46/117, 39.3% vs. 15/132, 11.4%, *p* < 0.001).

Moreover, those patients with at least partial response to PPIs were more frequently diagnosed with GERD when undertaking pH + impedance-monitoring examination irrespectively of the AET cut-off. When using the Lyon Consensus cut-off, a significant statistical correlation between GERD diagnosis and PPI responsiveness was observed (43/112 (38.4%) among responders to PPI compared with 26/137 (19.0%) of non-responders, *p* < 0.001). This correlation continued to exist in the case of using the older AET cut-off (54/112 (48.2%) among responders vs. 27/137 (19.7%) among non-responders, *p* < 0.001).

In the evidence supporting pH + impedance, as defined by the Lyon Consensus, positive SAP and/or SI were observed in 10/69 (14.5%) patients among those finally diagnosed with GERD using the Lyon Consensus criteria. Positive symptomatic indexes were not correlated to GERD diagnosis (*p* = 1), whereas abnormal MNBI was correlated to GERD diagnosis based on the Lyon Consensus criteria (16/63 (25.4%) vs. 22/186 (11.8%), *p* = 0.027).

In addition, the presence of endoscopic findings supportive of GERD diagnosis as stated by the Lyon Consensus (grade A and B esophagitis and/or the presence of hiatal hernia) was also significantly correlated with conclusive GERD on pH + impedance monitoring. The vast majority of the patients (31/42, 73.4%) with supportive endoscopic data (grade A and B esophagitis and/or the presence of hiatal hernia), as defined by the Lyon Consensus, were ultimately diagnosed with GERD by pH–impedance monitoring, in comparison to 29/130 (22.3%) of the patients without supportive endoscopic data, a difference which was statistically significant (*p* < 0.001).

In the multivariate analysis, however, only the existence of endoscopic findings supportive of GERD diagnosis as defined by the Lyon Consensus, a De Meester score of over 14.7, and the presence of typical GERD symptoms were correlated to GERD diagnosis (*p*-0.001, respectively) in terms of the criteria defined for pH + impedance monitoring (Table 3).

## 4. Discussion

Until recently, GERD was considered a clinical diagnosis based on the presence of typical symptoms. However, this old concept of the disease as a single clinical entity has been changed since a variety of symptoms have been considered putative reflux symptoms, leading to inaccurate diagnoses and the inappropriate use of medical therapies [17]. Thus, a definition of objective parameters, based on the available ambulatory esophageal monitoring, which can lead to a conclusive diagnosis or exclusion of GERD is highly needed. Indeed, the Lyon Consensus proposed that specific metrics on 24 h pH + impedance monitoring could truly discriminate patients with pathological GERD. It is suggested that 24 h pH–impedance monitoring off PPI is the optimal testing for those patients with indefinite GERD diagnosis. According to our findings, the recent changes proposed by the Lyon Consensus in the 24 h pH + impedance monitoring diagnostic criteria of GERD could establish a conclusive GERD diagnosis in only 25% of our population. It is of importance to state that this rate of diagnosis is significantly reduced compared with the one reported while using the previous Lyon criteria. We also observed this result in those patients with supportive but inconclusive evidence of GERD on endoscopy, as well as those with typical symptoms in the majority of whom a GERD diagnosis was confirmed.

It is well-known that GERD is a complex disease with a heterogeneous symptom profile and a multifaceted pathogenic basis that defies a simple diagnostic algorithm or categorical classification. A combined 24 h pH–impedance method that can detect all reflux episodes regardless of acidity is considered the gold standard for GERD diagnosis providing confirmatory evidence for pathological reflux. The primary outcome of reflux monitoring is AET, which is a continuous metric that could be assessed automatically, and it represents the most reproducible one [18]. Although AET is predictive of a good response to either medical or surgical treatment, a specific threshold value has not been proposed [19,20]. Most published studies used the cut-off of 4.2%, a value selected based on the evidence that a higher AET value proportionally correlates with the degree of reflux severity [21]. Using this cut-off, we found that almost one-third of our patients could meet the criteria for GERD diagnosis. Recently, the Lyon Consensus proposed that AET > 6% be considered clearly abnormal, whereas AET < 4% be considered definitely normal. The proposed AET cut-off of >6% might be considered more specific, and using it as the key outcome of the 24 h pH–impedance testing could potentially rule out GERD diagnosis in a substantial proportion of patients. Indeed, our data confirmed that less than 25% of our patients had conclusive evidence for pathological reflux when the new cut-off was administered. Our results are similar to those of a recent study that evaluated the use of novel impedance–pH parameters and their association with PPI response, in those patients with an inconclusive diagnosis of GERD according to the Lyon Consensus. The authors reported that only 62/233 patients (26.6%) had an AET value of >6% [22]. The same Italian study group further analyzed patients with proton pump inhibitor–refractory heartburn and stated that, among the patients examined off-therapy, abnormal AET values according to the Lyon Consensus criteria were observed in 23% of the study population, a percentage similar to ours [23]. It must, however, be stated that the researchers included those patients with only typical symptoms (heartburn) in comparison to our study, which included the patients with both typical and atypical symptoms. This may explain the increased rate of GERD diagnosis (37% off-PPI treatment) in the abovementioned study when also implementing the Lyon Consensus supportive evidence.

Additionally, our data point out that, in a great percentage of patients, PPIs are mistakenly prescribed, and the administration of firmer diagnostic criteria may lead to a reduction in PPI overuse [17]. Therefore, an investigation of a range of different diseases with distinctive pathogenesis and therapeutic management could be suggested. Neuromodulators or cognitive behavioral treatment, instead of PPIs, could be considered more appropriate therapies for those patients.

According to the Lyon Consensus, an AET value between 4% and 6% is considered inconclusive, and in this case, adjunctive outcome metrics should be used, in order to enhance the reliability of the diagnosis of the presence of GERD. The total NRE, reflux–symptom association, and novel metrics such as MNBI are among the proposed metrics that could be reviewed in the tracing. NRE > 80 per 24 h is considered abnormal and seems to be a useful tool in intermediate AET values, although data on the clinical relevance of an abnormal NRE are still controversial [24,25]. Reflux–symptom association analysis (SI and/or SAP) has a high degree of reproducibility, and both indexes are predictive of the success of medical and surgical anti-reflux therapy [26,27]. Moreover, these parameters are important for the diagnosis of reflux hypersensitivity and functional heartburn. We have to stress that these metrics are considered reliable enough, only when at least three symptom events occur during the monitoring [20]. A low MNBI is a useful tool in patients with inconclusive GERD and could unveil an indefinite GERD diagnosis in these patients. A value of <2292 ohms independently of AET predicts the response to anti-reflux therapy [28]. Our study showed that approximately 10% of our patients had inconclusive evidence of GERD. However, only a minority of them (13.7%) displayed evidence of pathological reflux based on the implementation of supportive pH–impedance metrics. Our findings are in contrast with those reported in a recent study showing a higher rate of inconclusive GERD diagnosis, as well as a greater number of definitive diagnoses of reflux disease when MNBI and post-reflux swallow-induced peristaltic wave (PSPW) indexes were used in those patients with inconclusive GERD [22]. This discrepancy could be due to the different study populations; we included those patients with both typical and atypical reflux symptoms, whereas Ribolsi et al. reviewed the tracings of those patients with only typical symptoms. Thus, further studies are needed in order to clarify this topic.

Trying to identify putative prognostic factors for GERD diagnosis, in the multivariate analysis, we found that the presence of typical GERD symptoms was strongly correlated to this diagnosis. As mentioned before, the diagnostic accuracy of the medical history, even if taken by an expert, does not exceed 70%, though our data point out that this diagnostic strategy is valid, especially when taking the great prevalence of GERD into consideration. Moreover, when interpreting our data in a reverse manner, this finding empowers the value of the criteria implemented by the Lyon Consensus for GERD diagnosis.

In the univariate analysis, we also showed that the treatment response to PPIs was closely correlated to GERD diagnosis, irrespective of the criteria used for GERD diagnosis, but this finding was not observed in the multivariate analysis. Older data have shown that patients with PPI responsiveness as well as those with typical GERD symptoms tend to have higher rates of treatment success when they undertake surgical treatments for GERD [29]. By strengthening the diagnostic criteria of pH–impedance monitoring in GERD diagnosis, fewer patients will have a clear indication for surgery. One could argue that the percentage of unsuccessful surgical management of GERD, in those patients diagnosed according to the older criteria, may have arisen due to GERD misdiagnosis.

Additionally, in the Lyon Consensus, it is stated that the existence of grade A or B esophagitis and/or hiatal hernia in upper GI endoscopy are supportive but not diagnostic lines of evidence of GERD. In our cohort, one out of the two patients showing supportive endoscopic findings of GERD as defined by the Lyon Consensus was finally diagnosed with GERD. Most importantly, the existence of the endoscopic supportive evidence of GERD was correlated in the multivariate analysis with GERD diagnosis with pH–impedance monitoring. Our finding is in agreement and further supports the statement adopted by the Lyon Consensus.

It must also be underlined that another significant finding of our study was that 63.4% of those patients who reported typical GERD symptoms while also showing at least partial response to PPIs and grade A or B esophagitis and/or hiatal hernia were finally diagnosed with GERD with pH–impedance monitoring. The presence of all these three prerequisites was strongly correlated to GERD diagnosis.

As far as we know, this is one of the first studies evaluating the diagnostic yield of pHmetry ± impedance using the Lyon Consensus criteria in a real-world setting. Our study population reflects the everyday clinical reality by including those patients presenting with typical or atypical esophageal symptoms who were evaluated with 24 h pH–impedance in order to establish a definitive GERD diagnosis. All the patients had received a preliminary treatment with PPIs indicating clinical practice and were responders or non-responders to PPIs. The main drawback of our study is its retrospective nature, even though data collection was prospectively performed. Another limitation tempering the strength of our study was the day-to-day variability of 24 h pH–impedance monitoring that could underestimate the rate of GERD diagnosis.

At this point, our main goal was to map the changes in GERD diagnosis by adopting the new, stricter, and well-defined criteria for GERD diagnosis proposed by the Lyon Consensus. A further study evaluating the impact of these new criteria on different treatment modalities is our intention. Keeping in mind the well-known disadvantages (cost, tolerance, and availability) of a pHmetry + impedance examination, the results of our correlation analysis could be useful in the better selection of the population in whom the study will be implemented.

In conclusion, our study showed that changes in the 24 h pH–impedance monitoring diagnostic criteria of GERD based on the Lyon Consensus could establish a conclusive GERD diagnosis in approximately 25% of our population. This rate represented a substantial reduction in GERD diagnosis compared with the one confirmed with the use of previously established criteria. The presence of typical symptoms of GERD, erosive esophagitis, and/or hiatal hernia was independently associated with the abnormal data of esophageal pH–impedance monitoring.

## Figures and Tables

**Figure 1 jcm-11-05383-f001:**
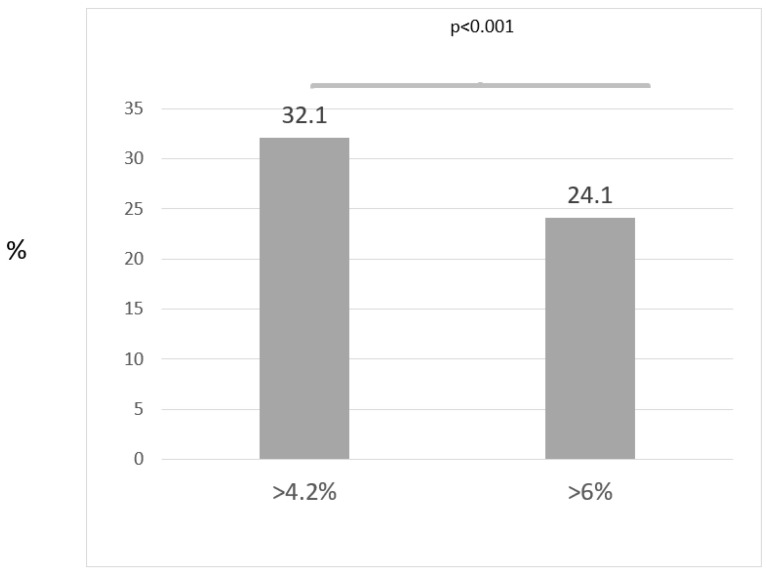
The rate of GERD diagnosis using different pathological AET cut-offs.

**Table 1 jcm-11-05383-t001:** Patients’ symptoms.

Patient Symptom	*n* (%)
Heartburn	121 (48.6)
Regurgitation	96 (38.6)
Dysphagia	15 (6.0)
Retrosternal pain	45 (18.1)
Extraesophageal manifestation	90 (36.1)

**Table 2 jcm-11-05383-t002:** PHmetry + impedance findings.

pHmetry + Impedance Examined Factors	*n* (%)
AET < 4%	147/249 (59%)
4 < AET < 6%	22/249 (8.9%)
AET > 6%	60/249 (24.1%)
Number of reflux episodes:	
over > 80 reflux	12/249 (4.8%)
40 < number of reflux episodes < 80	62/249 (24.9%)
<40	175/249 (70.3%)
De Meester score	
>14.7	74/249 (29.7%)
<14.7	175/249 (70.3%)
SI	
>50%	16/172 (9.3%)
<50%	156/172 (90.7%)
SAP	
<95%	142/172 (82.6%)
>95%	30/172 (17.4%)
MNBI	
<2292 Ohms	38/249 (15.3%)
>2292 Ohms	211/249 (84.7%)

AET: acid exposure time, SI: symptomatic index, SAP: symptom association probability, MNBI: mean nocturnal basal impedance.

**Table 3 jcm-11-05383-t003:** Multivariate analysis of factors correlated to GERD diagnosis.

	*p* (Univariate Analysis)	*p* (Multivariate Analysis)	Exp (B)
Sex	0.028	0.807	
Age	0.097	0.703	
BMI	0.015	0.170	
Presence of typical GERD symptoms	<0.001	0.036	0.281
PPI response	<0.001	0.775	
De Meester score > 14.7	<0.001	<0.001	12.336
MNBI * < 2292 Ohm	0.027	0.197	
Supportive endoscopic finding	<0.001	<0.001	9.963

MNBI *: mean nocturnal basal impedance.

## Data Availability

The data presented in this study are available on request from the corresponding author. All data retrieved by patients records, followed up in our hospitals’ outpatient clinic and were anonymously analysed. The study was conducted in compliance with the ethics principles for medical re-search involving human subjects as stated in the Declaration of Helsinki of the World Medical Association.

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
