# Peer review of "The Lyon Consensus Criteria for GERD Diagnosis in a Greek Population: The Clinical Impact and Changes in GERD Diagnosis in a Real-World, Retrospective Study"

_jcm, 2022, doi:10.3390/jcm11185383_

Round 1

Reviewer 1 Report

Dear authors!

I read with interest your manuscript “Validation of The Lyon Consensus criteria for GERD Diagnosis in a Greek population; a retrospective study” which is based on the results of original retrospective single-center trial. You stated the aim to investigate the diagnostic yield that implementation of Lyon consensus gives us in diagnosis of gastroesophageal reflux disease. I feel that the study brings new to the field and may be interesting for the readers. However, I have some concerns that may require your attention.

Major flaws.

First, I think there is a need for correction of statements. Diagnosis of gastroesophageal reflux disease is integral and should incorporate clinical and instrumental data. What we establish when we perform pH(impedance) study is revealing “abnormal” (or pathological) gastroesophageal reflux, but not GERD. Thus, the paragraph 2.3 should be revised. This also might impact the whole concept of the study (detection of abnormal data of esophageal pH-impedance data with the use of different criteria, that may support the presence of GERD).

Then, to measure the diagnostic yield (according to the aim), or “validate” the Lyon Consensus criteria (according to the title) it would be necessary to compare the results obtained with the use of past and the new criteria in subjects with “true” GERD - in those who experienced frequent typical symptoms for at least 3 months and had bothersome symptoms not less than 6 months, who did not use any medications that could lead to the development of symptoms and had no history of surgery that could result in abnormal gastrointestinal motility or reflux itself (it would be better, if these subjects had erosive esophagitis of grade greater than B to have the conclusive diagnosis according to the Lyon consensus). What is done is not a validation, but really a measure of impact of implementation of the diagnostic criteria provided by Lyon consensus in specific cohort of subjects who did not respond or had partial response to PPI treatment and had mild grades of esophagitis, and therefore had inconclusive criteria for GERD diagnosis. This is not less important and actual, but should be interpreted correctly. According to the mentioned above, should the title of the study be revised to reflect specificity of the group and different impact of the Lyon and Porto criteria?

According to the general requirements to the manuscripts by the ICMJE, the data of the biomedical studies should be reproducible, and the subjects’ group should be homogenous and clear. It is not so in the present study. Beside mentioned above lack of details that allowed to ensure the presence of GERD, the study group did not consist of non-responders to PPI treatment patients, and the definitions of non-response or partial response to PPIs used in the study are not provided. If your group consisted of consequent “real-world” subjects, this should be mentioned and discussed appropriately. According to the description of the study group, only subjects with no- or partial response to PPIs were enrolled. However, a response to PPIs is described in the results of correlation analysis - please, explain.

It is not clear, whether ph(impedance) studies underwent any selection. Please, indicate whether quality assessment of pH-impedance tracing was performed and whether selection criteria were applied. Patients flow and allocation chart may help to understand whether all the available records were enrolled to the analysis and if not, why they were excluded. The diagram of study design would also be appreciated. I would suggest that the data of the results were shown on a figure, if possible.

It is not clear, whether all subjects underwent pH-impedance examination or some of them had only data of esophageal pH-monitoring. Please, either revise the text (and mention correctly the name of the examination as esophageal multichannel intraluminal pH-impedance study) or indicate the number of subjects who had the data of esophageal pH-monitoring only.

The clinical data are poorly described. It is important not only to realize the number of GERD patients, but to make clear the data of correlation analysis provided in the paper. The frequency of symptoms (or the number of subjects who experience typical symptoms of GERD frequently) is important. Moreover, GERD has a number of extraesophageal manifestations, and according to the Montreal classification, some of extraesophageal manifestations has only possible relationship with gastroesophageal reflux.

Minor flaws.

In Materials and Methods, the name of manufacturer of pH-impedance probes is provided; however, this is not done for the recorder. Please, add this information as well.

As it is stated in the results, “esophagitis was present in 25/172 (14.5%) patients”. Please, indicate whether the data on different grades of esophagitis are provided in absolute or relative numbers (namely, n or %), as the format of “grade A: 16, B: 13” is confusing.

In table 2, please, provide the number of patients who had SAP > 95% (instead of 0.05).

Line 206: please, revise the SI/SAP to SI and/or SAP.

Lines 333-334 “In our cohort one out of two patients, showing any of the above, were finally diagnosed with GERD” – the phrase is ambiguous: there were 25 subjects with erosive esophagitis and 24 – with hiatal hernia.

Lines 235-239 “Thirty one -one of forty two (73.4%) patients…” – please, explain or revise.

Please, provide the details of the correlation analysis (correlation coefficients, Spearman rank R and P values).

Last, but not the least. What do the results of the study give us? This matter is only partially mentioned in the discussion. It is clear, that the number of abnormal results suggestive for GERD will be lower in case we make the diagnostic thresholds higher. How the results of correlation analysis may be interpreted and implemented to the clinical practice? What are the future directions for the research?

I hope that my comments help to make your manuscipt better.

Author Response

We wish to thank all three reviewers for their important and meaningful comments. The authors have tried to cope with all the comments made by the reviewers, which without any dough led to a better formed manuscript.  

Reviewer 1.

Major flaws.

First, I think there is a need for correction of statements. Diagnosis of gastroesophageal reflux disease is integral and should incorporate clinical and instrumental data. What we establish when we perform pH(impedance) study is revealing “abnormal” (or pathological) gastroesophageal reflux, but not GERD. Thus, the paragraph 2.3 should be revised. This also might impact the whole concept of the study (detection of abnormal data of esophageal pH-impedance data with the use of different criteria, that may support the presence of GERD).

Answer: GERD is changed to gastro esophageal reflux (as correctly mentioned by the reviewer) in paragraph 2.3. As all patients were symptomatic the existence of reflux in the ph-metry study combined with patients symptoms and adjunctive evidence sets the diagnosis of GERD as stated by the Lyon consensus. The aim of the study was to evaluate the Lyon consensus criteria and compare them to previous one.

Then, to measure the diagnostic yield (according to the aim), or “validate” the Lyon Consensus criteria (according to the title) it would be necessary to compare the results obtained with the use of past and the new criteria in subjects with “true” GERD - in those who experienced frequent typical symptoms for at least 3 months and had bothersome symptoms not less than 6 months, who did not use any medications that could lead to the development of symptoms and had no history of surgery that could result in abnormal gastrointestinal motility or reflux itself (it would be better, if these subjects had erosive esophagitis of grade greater than B to have the conclusive diagnosis according to the Lyon consensus). What is done is not a validation, but really a measure of impact of implementation of the diagnostic criteria provided by Lyon consensus in specific cohort of subjects who did not respond or had partial response to PPI treatment and had mild grades of esophagitis, and therefore had inconclusive criteria for GERD diagnosis. This is not less important and actual, but should be interpreted correctly. According to the mentioned above, should the title of the study be revised to reflect specificity of the group and different impact of the Lyon and Porto criteria?

Answer. Title was changed according to reviewers proposal to ‘’ The Lyon Consensus criteria for GERD Diagnosis in a Greek population; Clinical impact and changes in GERD diagnosis; a retrospective study.’’ Moreover, in Methods paragraph 1 (patients) changes have been made according to reviewers comments. All our study patients have long-lasting symptoms, patients with concomitant medications in which GERD development could be attributed were originally excluded as well as  patients with previous surgeries affecting gastrointestinal motility).

According to the general requirements to the manuscripts by the ICMJE, the data of the biomedical studies should be reproducible, and the subjects’ group should be homogenous and clear. It is not so in the present study. Beside mentioned above lack of details that allowed to ensure the presence of GERD, the study group did not consist of non-responders to PPI treatment patients, and the definitions of non-response or partial response to PPIs used in the study are not provided. If your group consisted of consequent “real-world” subjects, this should be mentioned and discussed appropriately. According to the description of the study group, only subjects with no- or partial response to PPIs were enrolled. However, a response to PPIs is described in the results of correlation analysis - please, explain.

Answer: As stated in in method section paragraph 1 and in the discussion part, our group of patients indeed consisted of real world population .  Our study included patients who  were partial or non responders  to PPIs as well as to patients who were previously PPi responder (‘as well as responsive to PPIs but requiring 24-hrs pH-impedance monitoring in order to confirm GERD diagnosis in the setting of pre-surgical control for antireflux surgery.’’). ’A definition of complete response was added in the same paragraph

It is not clear, whether ph(impedance) studies underwent any selection. Please, indicate whether quality assessment of pH-impedance tracing was performed and whether selection criteria were applied. Patients flow and allocation chart may help to understand whether all the available records were enrolled to the analysis and if not, why they were excluded. The diagram of study design would also be appreciated. I would suggest that the data of the results were shown on a figure, if possible.

Answer  : Accoring to our center policy during the study period, all patients underwent ph-metry and impendence  therefore no selection was done.

It is not clear, whether all subjects underwent pH-impedance examination or some of them had only data of esophageal pH-monitoring. Please, either revise the text (and mention correctly the name of the examination as esophageal multichannel intraluminal pH-impedance study) or indicate the number of subjects who had the data of esophageal pH-monitoring only.

Answer Accoring to our center policy during the study period, all patients underwent ph-metry and impendence, therefore no selection was done, as stated above. The ± was changed to + in order not the readers to be confused.

The clinical data are poorly described. It is important not only to realize the number of GERD patients, but to make clear the data of correlation analysis provided in the paper. The frequency of symptoms (or the number of subjects who experience typical symptoms of GERD frequently) is important. Moreover, GERD has a number of extraesophageal manifestations, and according to the Montreal classification, some of extraesophageal manifestations has only possible relationship with gastroesophageal reflux.

Answer: In our study only patients with chronic cough, laryngitis and asthma were included. All of three considered as extraesophageal manifestations of GERD (according also to the latest AGA position statement). In order to be more analytic, a statement was added in the method section (paragraph: patients) as also in the results part.

Minor flaws.

In Materials and Methods, the name of manufacturer of pH-impedance probes is provided; however, this is not done for the recordersPlease, add this information as well.

Answer: It was  added ( Ohmega Portable 24h Impedance & PH system, MMS, Enschede, The Netherland)

As it is stated in the results, “esophagitis was present in 25/172 (14.5%) patients”. Please, indicate whether the data on different grades of esophagitis are provided in absolute or relative numbers (namely, n or %), as the format of “grade A: 16, B: 13” is confusing.

Answer: It was changed to absolute numbers

In table 2, please, provide the number of patients who had SAP > 95% (instead of 0.05).

Answer: the number is 30/172 as provided in the second column of the table 2. 0.05 was changed to >95% as requested

Line 206: please, revise the SI/SAP to SI and/or SAP.

Answer: corrected as requested

Lines 333-334 “In our cohort one out of two patients, showing any of the above, were finally diagnosed with GERD” – the phrase is ambiguous: there were 25 subjects with erosive esophagitis and 24 – with hiatal hernia.

Answer: The phrase was changed to ‘’showing supportive endoscopic findings of GERD as defined by the Lyon consensus’’ in order to clarify the meaning of the phrase which is that our study comes to agreement with the Lyon consensus statement which supports the use of grade A and B esophagitis as also hiatal hernia as supportive but not conclusive evidences of GERD.

Lines 235-239 “Thirty one -one of forty two (73.4%) patients…” – please, explain or revise.

Answer: The phrase was changed to ‘’ Thirty one -one of forty two (73.4%) patients with supportive endoscopic data (grade A and B esophagitis and/or presence of hiatal hernia) as defined by the Lyon consensus’’ so as not be misinterpreted.

Please, provide the details of the correlation analysis (correlation coefficients, Spearman rank R and P values).

Answer: No spearman analysis was used in the current paper.

Last, but not the least. What do the results of the study give us? This matter is only partially mentioned in the discussion. It is clear, that the number of abnormal results suggestive for GERD will be lower in case we make the diagnostic thresholds higher. How the results of correlation analysis may be interpreted and implemented to the clinical practice? What are the future directions for the research?

Answer:  At this point, our main goal was to map the changes in GERD diagnosis by adopting the the new, more strict and well defined criteria for GERD diagnosis proposed by the Lyon consensus. A further study evaluating the impact of these new criteria to different treatment modalities is in our intention. Keeping in mind the well-known disadvantages (cost, tolerance, and availability) of pHmetry+impedance examination, the results of our correlation analysis could be useful in the better selection of population in whom the study will be implemented.

Reviewer 2 Report

Summary

The article entitled "Validation of The Lyon Consensus criteria for GERD Diagnosis 2 in a Greek population; a retrospective study" is a retrospective study aimed to investigate investigate the diagnostic yield of pHmetry ± impedance using the Lyon consensus criteria. The main result are that endoscopic findings supportive of 38 GERD diagnosis based on Lyon consensus were present in 42/185 (22.7%) subjects, whereas an AET>6% was observed in 60/249 39 (24.1%) cases. Thus, GERD diagnosis according to the Lyon Consensus criteria was set in 63/249 (25.3%) patients; a rate significantly lower compared to the rate observed by implementing the older criteria 41 (32.1%).

The study is interesting and original. I have no major concerns. My minor concerns are:

  1. There are some language mistakes throughout the manuscript (i.e. line 78 “trail”, line 76 “;”, etc.). Please, correct them
  2. Please, clarify which “various extra-esophageal symptoms” have been considered and included
  3. I would add in the title that population studied was composed by subjects with partial/lack of response to medical therapy. Likewise, the low rate of GERD diagnosis should be regarded and discussed based on this point (i.e. it is well known that a minority of patients with lack of response to medical treatment , really have GERD). Please, amend the discussion to reflect this point of view
  4. SI and SAP should be calculated for typical symptoms and cough. Their value in other atypical symptoms such as belching or hoarseness is controversial. Please, clarify this issue accordingly
  5. It should be interesting to know the rate of response to medical or surgical therapy among patients with a Lyon Consensus diagnosis of GERD versus those who were not diagnosed with GERD. I strongly suggest to add this data
  6. The lack of PSPW calculation whit the explanation of what it is (with appropriate references) should be added among the limitations of the study
  7. This is not the first study applying the Lyon Consensus criteria, and investigating their validation (i.e. Aliment Pharmacol Ther. 2022 Jun;55(11):1423-1430). Please, cite these papers and comment in the discussion section

Author Response

We wish to thank all three reviewers for their important and meaningful comments. The authors have tried to cope with all the comments made by the reviewers, which without any dough led to a better formed manuscript.  

There are some language mistakes throughout the manuscript (i.e. line 78 “trail”, line 76 “;”, etc.). Please, correct them

Answer: An effort to correct any grammar or syntactic errors was made

Please, clarify which “various extra-esophageal symptoms” have been considered and included

Answer: In our study only patients with  chronic cough, laryngitis and asthma were included. All of three considered as extraesophageal manifestations of GERD (according also to the latest AGA position statement). In order to be more analytic, a statement was added in the method section (paragraph: patients) as also in the results part.

I would add in the title that population studied was composed by subjects with partial/lack of response to medical therapy. Likewise, the low rate of GERD diagnosis should be regarded and discussed based on this point (i.e. it is well known that a minority of patients with lack of response to medical treatment , really have GERD). Please, amend the discussion to reflect this point of view

Answer: Οur group of patients consisted of real world population and not only of ppi refractory patients as also analysed in method section paragraph 1.  Our study included also patients who were previously PPi responder (‘as well as responsive to PPIs but requiring 24-hrs pH-impedance monitoring in order to confirm GERD diagnosis in the setting of pre-surgical control for antireflux surgery.’’).Though the title was changed in order to underline the fact that our work is a real world study. A definition of complete response was added in the same paragraph

SI and SAP should be calculated for typical symptoms and cough. Their value in other atypical symptoms such as belching or hoarseness is controversial. Please, clarify this issue accordingly

Answer: Symptoms assessed were heartburn, regurgitation, chest pain, cough or belching as reported in the original paper of the Lyon consensus. As proposed by the reviewer in the method section, paragraph ph-metry + impedance a definition of assessed was introduced.

It should be interesting to know the rate of response to medical or surgical therapy among patients with a Lyon Consensus diagnosis of GERD versus those who were not diagnosed with GERD. I strongly suggest to add this data

Answer: Unfortunately there are no consistent data about the follow up of patients treated medically or surgically available.  Our scope is to perform a further study in order to evaluate the impact of these new criteria to different treatment modalities

The lack of PSPW calculation whit the explanation of what it is (with appropriate references) should be added among the limitations of the study

Answer

A comment was added in the method section, page 8, paragraph 1: ’’ PSWP, a novel PH-metry metric which may in the future add to GERD diagnosis was not measured in our study due to lack of resources’’

This is not the first study applying the Lyon Consensus criteria, and investigating their validation (i.e. Aliment Pharmacol Ther. 2022 Jun;55(11):1423-1430). Please, cite these papers and comment in the discussion section

Answer: A comment of the abovementioned study was added in the discussion part (page 18 par 1)

Reviewer 3 Report

The article compares the outcome of pH-metry and impedance evaluation according to new and old criteria. The authors conclude that according to the Lyon criteria GERD diagnosis can be stated less often. They also show that the additional metrics such as SI, SAP, MNBI and NRE, rarely specify the diagnosis in the group of patients with AET between 4 and 6 %

The study is important and generally well conducted but in my opinion some points are not sufficiently discussed.

So far pH-metry with impedance was considered gold standard of GERD diagnosis. Now if the patient has 4-6% AET  in most cases this gold standard cannot state if the patient has GERD or not.  So what can?

What should the practitioner propose to the patient with inconclusive pH-metry result? What measures can the practitioner take to treat those patients in a  tailored manner?

Although the results of this study were easy to prognose the study is important because it shows how much has to be clarified in the future.

How this new criteria affected the diagnosis in patients with non-acidic reflux in the study group?

Other questions:

Is it possible to state how long was the break between the last PPI dose and the examination?

The table with univariate correlations would make the reading easier.

How many patients with typical symptoms , esophagitis A and partial response to PPI was not diagnosed with GERD  according to the new criteria?

Minor comments:

Line 39

Abbreviation AET is not explained.

Line 56

The word “caustic” is unsuitable because not only caustic reflux cause symptoms.

Line 118

The word “overnight is redundant.

Line 260

Shouldn’t the word “inconclusive” be rather conclusive?

Line 354

Examined population. In the present shape it may seem that general population.

Author Response

We wish to thank all three reviewers for their important and meaningful comments. The authors have tried to cope with all the comments made by the reviewers, which without any dough led to a better formed manuscript.  

So far pH-metry with impedance was considered gold standard of GERD diagnosis. Now if the patient has 4-6% AET  in most cases this gold standard cannot state if the patient has GERD or not.  So what can?

Answer: Answer is given by the Lyon consensus group by the addition of supportive/adjunctive criteria.  It is not the scope of this import new criteria but to evaluate the ones formed by the Lyon consensus

What should the practitioner propose to the patient with inconclusive pH-metry result? What measures can the practitioner take to treat those patients in a tailored manner?

Answer: :  At this point, our main goal was to map the changes in GERD diagnosis by adopting the new, more strict and well defined criteria for GERD diagnosis proposed by the Lyon consensus. A further study evaluating the impact of these new criteria to different treatment modalities is in our intention.

Is it possible to state how long was the break between the last PPI dose and the examination?

Answer: All patients have stopped PPi at least 2 weeks before the examination according to our center policy. A sentence was added in the method section, paragraph: patients

The table with univariate correlations would make the reading easier.

Answer: Table 3 was reformed in order to include the univariate analysis also.

How many patients with typical symptoms , esophagitis A and partial response to PPI was not diagnosed with GERD  according to the new criteria?

Answer: As stated in the result part only 13 patients among 172 patients with available endoscopic data had grade A esophagitis. Among them only 3 had both partial response to PPis and typical esophageal symptoms. Due to the small number of patients belonging to this certain category, the data are not given in the results part, whereas no comparative analysis could be executed.

Minor comments:

Line 39

Abbreviation AET is not explained.

Answer: AET is explained in page 5 line 6

Line 56

The word “caustic” is unsuitable because not only caustic reflux cause symptoms.

Answer: Was changed to burning pain

Line 118

The word “overnight is redundant.

Answer: word deleted as not needed

Line 260

Shouldn’t the word “inconclusive” be rather conclusive?

Answer: was changed to ‘’ supportive but not conclusive’’

Round 2

Reviewer 1 Report

I read the authors' response to my comments and the revised version of the manuscript. The paper has been significantly improved, but still requires further polishing. Some of the comments remained unanswered. 

Please, revise the text of the study aim according to the previous comment and your response. Please, pay attention to the correct name of the procedure. As all the participants underwent combined (pH and impedance) recordings, it would be better to use "and" or "+" instead of "+/-". 

According to the quality control of the recordings: this information is required to ensure that tracings lasted sufficient time, there was no displacement of probe and no artifact influenced the results. Please, add the statement to the Methods section. 

Please, provide the definition of partial responders and non-responders to PPI treatment. 

Please, ensure that uniformed abbreviation of proton pump inhibitors (PPI) is correctly used (not PPis, or else) throughout the text.

Lines 246-251: the sentence is still unclear. Please, rephrase the part of "Thirty one -one of forty two (73.4%) patients..." as it remains confusing. 

Lines 379-385: please, move this information upper, to the discussion. 

Please, revise the conclusions in the abstract and in the manuscript's body according to the study aim and the results. Please, specify the population used in the study here.  I would suggest to provide the value of reduction of the number of subjects with established diagnosis of GERD, rather than just a proportion of subjects with established diagnosis of GERD in the studied population, as it is more important. Could be "Presence of typical symptoms of GERD, erosive esophagitis and/or hiatal hernia were independently associated with abnormal data of esophageal pH-impedance-monitoring" better than provided in conclusions (lines 377-379)? 

Author Response

We wish to thank the reviewer for the important and meaningful comments. The authors have tried to cope with all the comments made by the reviewers, which without any dough led to a better formed manuscript.  .

Answers to reviewers comment

Please, revise the text of the study aim according to the previous comment and your response. Please, pay attention to the correct name of the procedure. As all the participants underwent combined (pH and impedance) recordings, it would be better to use "and" or "+" instead of "+/-".  

The aim section was revised in order to adopt to the reviewers comments ‘’ The aim of our study was to investigate the diagnostic yield of pHmetry + impedance using the Lyon consensus criteria in a real world study and to evaluate possible changes in the rate of GERD diagnosis with the adaptation of those new diagnostic criteria among patients, partially or not at all responders to PPIs without conclusive endoscopic evidence of GERD and patients with a previous GERD diagnosis  and a complete response to PPIs who were candidates for surgical treatments. ‘’

The symbol +/- was revised in +

According to the quality control of the recordings: this information is required to ensure that tracings lasted sufficient time, there was no displacement of probe and no artifact influenced the results. Please, add the statement to the Methods section.

In the method section a statement was added : Only pH-impedance studies in which the probe remained in place for at least 16 hrs were considered valid for analysis. Artifact recognized by the operator were manually excluded from the analysis

Please, provide the definition of partial responders and non-responders to PPI treatment.

Definition of a partial and no response to PPI was added in the method section:

Partial response was defined as an amelioration of patient symptoms but without com-plete disappearance of them and no response was defined as no change to patient symptoms. 

Please, ensure that uniformed abbreviation of proton pump inhibitors (PPI) is correctly used (not PPis, or else) throughout the text.

Revised universally in the text.

Lines 246-251: the sentence is still unclear. Please, rephrase the part of "Thirty one -one of forty two (73.4%) patients..." as it remains confusing.

The sentence was changed to: The vast majority of patients (31/42, 73.4%) with supportive endoscopic data (grade A and B esophagitis and/or presence of hiatal hernia) as defined by the Lyon consensus, were ultimately diagnosed with GERD by pH- impedance monitoring, in comparison to 29/130 (22.3%) of patients without supportive endoscopic data, a difference which was statistical significant (p<0.001).

Lines 379-385: please, move this information upper, to the discussion.

Answer: the statement  ‘’At this point, our main goal was to map the changes in GERD diagnosis by adopting the the new, more strict and well defined criteria for GERD diagnosis proposed by the Lyon consensus. A further study evaluating the impact of these new criteria to different treat-ment modalities is in our intention. Keeping in mind the well-known disadvantages (cost, tolerance, and availability) of pHmetry+impedance examination, the results of our correlation analysis could be useful in the better selection of population in whom the study will be implemented’’ was transported upper in the discussion as requested.

Please, revise the conclusions in the abstract and in the manuscript's body according to the study aim and the results. Please, specify the population used in the study here.  I would suggest to provide the value of reduction of the number of subjects with established diagnosis of GERD, rather than just a proportion of subjects with established diagnosis of GERD in the studied population, as it is more important. Could be "Presence of typical symptoms of GERD, erosive esophagitis and/or hiatal hernia were independently associated with abnormal data of esophageal pH-impedance-monitoring" better than provided in conclusions (lines 377-379)?

Answer: In the abstract the phrase:real world study was added in order to reflect the study population. More analytic description has been given in the aim part in the introduction as stated above. In the conclusion section the phrase ‘’Moreover, presence of typical symptoms, as well as existence of endoscopic supportive ev-idence of GERD was independently correlated with GERD diagnosis by pH-impedance-monitoring’’ was changer to Presence of typical symptoms of GERD, erosive esophagitis and/or hiatal hernia were independently associated with abnormal data of esophageal pH-impedance-monitoring as proposed by the reviewer.

Reviewer 2 Report

I thing the authors made a good job in revising the paper

Author Response

A spell check was redone and minor linguistic changes executed